Efficacy of different dietary therapy strategies in active pediatric Crohn’s disease: a systematic review and network meta-analysis

Ma Jiaze 1 2 3
Chong Jinchen 1 2 3
Qiu Zhengxi 1 2 3
Wang Yuji 1 2 3
Chen Tuo 4 chentuoysh@163.com
Chen Yugen 1 2 5 yugen.chen@njucm.edu.cn
1 The Affiliated Hospital of Nanjing University of Chinese Medicine, Nanjing University of Chinese Medicine , Nanjing , China
2 Jiangsu Province Key Laboratory of Tumor Systems Biology and Chinese Medicine, The Affiliated Hospital of Nanjing University of Chinese Medicine, Nanjing University of Chinese Medicine , Nanjing , China
3 No. 1 Clinical Medical College, Nanjing University of Chinese Medicine , Nanjing , China
4 Department of General Surgery, Affiliated Hospital of Yangzhou University, Yangzhou University , Yangzhou , China
5 Jiangsu Collaborative Innovation Center of Chinese Medicine in Prevention and Treatment of Tumor, The Affiliated Hospital of Nanjing University of Chinese Medicine, Nanjing University of Chinese Medicine , Nanjing , China
Wang Jincheng
Electronic publication date: 2024 Dec 13
Publication date: 2024
Volume: 12
Electronic Location ID: e18692
Received 2024 Oct 25; Accepted 2024 Nov 20
Copyright: © 2024 Ma et al.
Copyright year: 2024
Copyright holder: Ma et al.
License: This is an open access article distributed under the terms of the Creative Commons Attribution License, which permits unrestricted use, distribution, reproduction and adaptation in any medium and for any purpose provided that it is properly attributed. For attribution, the original author(s), title, publication source (PeerJ) and either DOI or URL of the article must be cited.
License URL: https://creativecommons.org/licenses/by/4.0/

Keywords: Crohn’s disease, Inflammatory bowel disease, Dietary therapy strategies, Systematic review, Meta-analysis, Children

Funding: National Natural Science Foundation of China 82305242 Jiangsu Funding Program 2023ZB483 Nanjing Postdoctoral Research Grant Program Yangzhou Key Research and Development Project No YZ2023076 This work was supported by the National Natural Science Foundation of China (No. 82305242), the Jiangsu Funding Program for Excellent Postdoctoral Talent (No. 2023ZB483), Nanjing Postdoctoral Research Grant Program, and the Yangzhou Key Research and Development Project (Social Development, No. YZ2023076). The funders had no role in study design, data collection and analysis, decision to publish, or preparation of the manuscript.

==============================
Background

Dietary therapy strategies play an important role in the treatment of pediatric patients with Crohn’s disease (CD), but the relative efficacy of different dietary therapy strategies for Crohn’s remission is unknown. This study aims to compare the effectiveness and tolerance of these dietary therapy strategies for active pediatric CD.

Methods

We searched the medical literature up to August 30, 2024 to identify randomized controlled trials (RCTs) of dietary therapy strategies for pediatric CD. The primary outcomes were clinical remission rate and tolerance, secondary outcomes included differences between pre- and post-treatment levels of albumin, C-reactive protein (CRP), and fecal calprotectin levels. A network meta-analysis (NMA) was performed by using the frequentist model. For binary outcome variables and continuous outcome variables, odds ratios (OR) and mean differences (MD) with corresponding 95% confidence intervals (CI) were utilized, respectively. The ranking of dietary therapy strategies was determined based on the surface under the cumulative ranking area (SUCRA) for each comparison analyzed.

Results

Overall, 14 studies involving 564 participants were included. In terms of clinical remission rate, the partial enteral nutrition (PEN) plus Crohn’s disease exclusion diet (PEN+CDED) (OR = 7.86, 95% CI [1.85–33.40]) and exclusive enteral nutrition (EEN) (OR = 3.74, 95% CI [1.30–10.76]) exhibited significant superiority over PEN alone. The tolerance of PEN+CDED was significantly higher than that of EEN (OR = 0.07, 95% CI [0.01–0.61]). According to the surface under the cumulative ranking area (SUCRA) values, the PEN+CDED intervention (90.5%) achieved the highest ranking in clinical remission rate. In terms of tolerance, PEN+CDED ranked first (88.0%), while EEN ranked last (16.3%).

Conclusions

In conclusion, PEN+CDED was associated with the highest clinical remission rate and tolerance among the various dietary therapy strategies evaluated. Despite limitations in the studies, this systematic review provides evidence that PEN+CDED can be used as an alternative treatment to exclusive enteral nutrition and is more suitable for long-term management in children.

Introduction

Crohn’s disease (CD) is a subtype of inflammatory bowel disease characterized by recurrent inflammation affecting both intestinal and extraintestinal manifestations (Kaplan, 2015). It often leads to significant weight loss, malnutrition, and quality of life decline. It is noteworthy that the incidence of pediatric CD is escalating worldwide, and up to 25% of cases in adults have been diagnosed in childhood (Benchimol et al., 2017; Rosen, Dhawan & Saeed, 2015; Turunen et al., 2009). Due to the unique aspects of growth and development in children, the treatment strategies for pediatric CD differ from those for adults.

Although immune suppressive therapy has been as a primary treatment for CD, it often results in increased economic burden and risk of infections and cancer (Olén et al., 2019; Engel et al., 2019; He, Yan & Wu, 2023). Thus, it is necessary to explore safe and effective alternative therapy for pediatric patients with CD. Dietary therapy has been found to have a great influence on pediatric CD, which are related to plays a crucial role in regulating intestinal microbiota, metabolism, and immune responses (Martinez-Medina et al., 2014; Levine, Sigall Boneh & Wine, 2018). Moreover, it is better than drugs because it does not increase immunosuppression and has no obvious side effects. Nowadays, dietary therapy has been the first choice for inducing remission in pediatric CD (Jang, Kang & Choe, 2019; Colombel et al., 2017).

Dietary therapy for pediatric CD involves in several strategies including exclusive enteral nutrition (EEN), partial enteral nutrition (PEN), and elimination diets (ED) (González-Torres et al., 2022). EEN is the first line therapy for mild-to-moderate pediatric CD, but its treatment is difficult to implement due to poor patient tolerability (van Rheenen et al., 2020; Ruemmele et al., 2014). Its unacceptability has led to the exploration of other strategies. PEN is characterized with allowing regular food and liquid formulations (Pigneur & Ruemmele, 2019). ED alleviate pediatric CD by eliminating exacerbating dietary factors to promote self-healing, which include the Crohn’s Disease Exclusion Diet (CDED), the Anti-inflammatory diet for Crohn’s disease (AID-CD) (Verburgt et al., 2021).

Although those dietary therapy strategies have been shown to be beneficial treatment for pediatric CD in several randomized controlled trials, the optimal dietary therapy strategies for these patients have not been confirmed (Jagt et al., 2023). Accordingly, we conducted the systematic review and network meta-analysis of to figure out the most appropriate dietary therapy strategies for pediatric CD.

Methods

The protocol of network meta-analysis (NMA) was registered with PROSPERO (CRD42023430213), and followed the PRISMA Extension Statement for Reporting of Systematic Reviews Incorporating Network Meta-analyses (Hutton et al., 2015).

Search strategy

We completed a thorough search of PubMed, Embase, and Cochrane Library on May 1, 2023, and ran it again on August 30, 2024, without language or time restrictions. The search terms “diet therapy” and “nutrition therapy” were combined using “OR”, the search terms “inflammatory bowel diseases” and “Crohn’s disease” were combined using “OR”, and the search terms “pediatric” and “child” were combined using “OR”. These three groups were then combined using “AND”. Medical subject headings (MeSH) were used to search the online databases. For example, searching in PubMed: ((pediatric[MeSH]) OR child[MeSH])) AND ((diet therapy[MeSH]) OR nutrition therapy[MeSH])) AND ((Crohn disease[MeSH]) OR inflammatory bowel diseases[MeSH]). We also reviewed reference lists of retrieved studies and relevant reviews to identify any potentially missed trials. In addition, we searched clinicaltrials.gov for unpublished trials or supplementary data. The detailed search strategy is available in Table S1.

Selection criteria

The two authors (Jiaze Ma, Jinchen Chong) independently removed duplicate articles and then identified and reviewed each article based on the title, abstract, and full text. In cases where consensus was not reached, Tuo Chen and Yugen Chen acted as referees. Finally, articles were extracted based on the following criteria: (1) were randomized controlled trials (RCTs), (2) the study population consisted of pediatric patients (under 18 years old) with active Crohn’s disease, exclusion criteria included recent use of steroids, recent initiation or dose adjustment for immunomodulators, past or current biologics use, primary colonic disease with significant rectal involvement, or active perianal disease, (3) the trials tested the efficacy of dietary therapy strategies for CD remission and compared them either to each other or to other treatment modalities, (4) RCTs with an intervention duration of 6 weeks or longer (induction phase of treatment only), and if efficacy data were provided before the crossover in a crossover trial, they were eligible for inclusion, and (5) patients received dietary therapy alone during the induction phase without concurrent use of steroids or other therapies.

If a study reported data related to the study factors at multiple intervention times, the longest or the last intervention time was considered. Observational and descriptive analysis studies, animal articles, case series/reports, clinical guidelines, reviews and editorials, as well as studies with duplicate or uncertain data for which no feedback was received after contacting the corresponding authors, were excluded from our study.

Data extraction and quality assessment

The two authors independently extracted data from each included study and entered it into Microsoft Excel spreadsheets. Any discrepancies between the two independent reviewers were resolved through discussion. If necessary, a third author provided arbitration. The primary outcome measure extracted was clinical remission rate (percentage of patients who achieved clinical remission defined as pediatric Crohn’s disease activity index (PCDAI) less than 10 after treatment) (Yu, Chen & Chen, 2019) and tolerance (defined as the patient’s withdrawal from the study due to refusal to continue the dietary therapy strategies) (Levine et al., 2019), secondary outcome measures differences between pre- and post-treatment levels of albumin, C-reactive protein, and fecal calprotectin among included studies. After the two independent reviewers reviewed the relevant articles, additional information such as author names, country, publication dates, participant numbers, and patient characteristics (including age, sex), intervention details (type of intervention, intervention time).

The potential risk of bias was evaluated by the two authors independently using the Cochrane Risk of Bias Tool, specifically designed for randomized controlled trials (Higgins et al., 2011).

Data synthesis and statistical analysis

According to the reporting guidelines outlined in the PRISMA extension statement for network meta-analysis, we performed the network meta-analysis using frequency-based models in STATA version 17.0 (Stata Corp., Tempe, AZ, USA) (Hutton et al., 2015). For binary outcome variables and continuous outcome variables, odds ratios (OR) and mean differences (MD) with corresponding 95% confidence intervals (CI) were utilized, respectively. If continuous data were presented as baseline and final mean values and standard deviations (SD), statistical algorithms were employed to calculate the mean differences and SD of the respective variables. Additionally, we created a table where the data within each cell represented the OR/MD values and 95% CI for comparisons between corresponding row and column treatment measures, indicating the results’ statistical significance when the 95% CI did not include 1/0. A network diagram was generated to visually represent all treatment comparisons, with the size of the nodes and connections reflecting the number of studies and the number of participants, respectively.

In studies that included both direct and indirect comparisons, we utilized the loop-specific approach to evaluate the presence of inconsistency. Minimal inconsistency was defined as the 95% CI of the inconsistency factor (IF) including 0. By employing network meta-analysis, we were able to rank the dietary therapy strategies using the surface under the cumulative ranking area (SUCRA), which provides a probability-based measure. Higher SUCRA values indicated a greater likelihood of being the optimal treatment strategy, offering valuable insights for clinical decision-making (Salanti, 2012). If sufficient studies (10 or more) were included in an analysis, we generated a comparison-adjusted funnel plot to identify potential publication bias.

Results

Literature research and characteristic of studies

Based on our search strategy, we initially identified a total of 2,800 relevant studies. After screening the titles/abstracts and removing duplicates, we obtained the full texts of 69 potentially eligible studies. Ultimately, 14 RCTs were included in our analysis (Kowalska-Duplaga et al., 2019; Thomas, Taylor & Miller, 1993; Kierkuś et al., 2013; Terrin et al., 2002; Luo et al., 2017; Pigneur et al., 2019; Papadopoulou et al., 1995; Hart et al., 2020; Borrelli et al., 2006; Johnson et al., 2006; Levine et al., 2019; Lee et al., 2015; Urlep et al., 2020; Ruuska et al., 1994). The systematic literature search and study selection process are visually depicted in Fig 1. These included studies were published between 1993 and 2023, and collectively involved a total of 564 patients. The dietary therapy strategies consisted of EEN (n = 10), PEN (n = 2), PEN+CDED (n = 1), PEN+AID-CD (n = 1). The characteristics of the included 14 RCTs are summarized in Table S2.

Figure 1 Search flow diagram.

The search flow diagram summarizes the search, screening, retrieval, and appraisal of articles finally included in the network meta-analysis

Quality assessment

A total of 10 RCTs demonstrated a low risk of bias in terms of inadequate sequence generation. Half of the studies lacked information on whether allocation was concealed, we assessed them as having an unclear risk of bias. All trials had a high risk of patient blinding, because dietary therapy strategies are almost impossible to be blinded. Four trials lacked blinding in outcome assessments, suggesting a potential risk of bias in outcome measurement. It is noteworthy that all RCTs included in this analysis demonstrated a low risk of selective reporting bias and incomplete outcome data, ensuring the reliability and completeness of the reported findings. Finally, in terms of other bias, five studies could not exclude other sources of bias because they lacked conflict of interest statement and were therefore evaluated as unclear. Detailed assessments of bias can be found in Fig. S1.

Primary outcome

Clinical remission rate

Clinical remission rates were reported in 11 RCTs with 467 pediatric CD. A total of 213 patients received various dietary therapy strategies (Levine et al., 2019; Kierkuś et al., 2013; Terrin et al., 2002; Luo et al., 2017; Papadopoulou et al., 1995; Hart et al., 2020; Borrelli et al., 2006; Johnson et al., 2006; Lee et al., 2015; Urlep et al., 2020; Ruuska et al., 1994). The network evidence plot was presented in Fig. 2A. The combined data showed good consistency between direct and indirect comparisons (IF = 1.07, 95% CI [0–4.10]) (Fig. S2). The comparison-adjusted funnel plot displayed no significant evidence of small-study effects within the included studies (Fig. S12A). Based on the SUCRA values (Fig. S3), PEN+CDED (SUCRA 90.5%) was the most effective dietary therapy strategies for clinical remission rate, followed by EEN (SUCRA 62.0%), PEN+AID-CD (SUCRA 57.6%). PEN (SUCRA 6.1%) was the least effective. The comparative forest plot was presented in Fig. S4. Table 1 summarized the NMA comparison results for the clinical remission rate, indicating that PEN+CDED (OR = 7.86, 95% CI [1.85–33.40]) and EEN (OR = 3.74, 95% CI [1.30–10.76]) were significantly more effective than PEN.

Figure 2 Network map.

(A) Network map for clinical remission rate. (B) Network map for tolerance. (C) Network map for CRP. (D) Network map for albumin. (E) Network map for fecal calprotectin. EEN, exclusive enteral nutrition; PEN, partial enteral nutrition; AID-CD, anti-inflammatory diet for Crohn’s disease; CDED, Crohn’s disease exclusion diet.

Table 1 Network meta-analysis comparisons for clinical remission rate.

PEN+CDED						
2.10 (0.78, 5.65)	EEN					
2.10 (0.19, 22.73)	1.00 (0.11, 8.73)	PEN+AID-CD				
2.35 (0.56, 9.94)	1.12 (0.39, 3.19)	1.12 (0.10, 12.43)	Infliximab			
3.89 (1.15, 13.11)	1.85 (0.92, 3.75)	1.85 (0.19, 18.08)	1.65 (0.47, 5.85)	Corticosteroids		
7.86 (1.85, 33.40)	3.74 (1.30, 10.76)	3.74 (0.34, 41.68)	3.34 (1.13, 9.90)	2.02 (0.57, 7.19)	PEN	
Notes:

ORs higher than 1 favour the column-defining treatment.

ORs lower than 1 favour the row-defining treatment.

Significant results are in bold.

EEN, exclusive enteral nutrition; PEN, partial enteral nutrition; AID-CD, anti-inflammatory diet for Crohn’s disease; CDED, Crohn’s disease exclusion diet.

Tolerance

A total of seven RCTs, involving 345 patients, reported the tolerance, with a total of 243 patients receiving dietary therapy strategies (Levine et al., 2019; Kierkuś et al., 2013; Luo et al., 2017; Borrelli et al., 2006; Johnson et al., 2006; Lee et al., 2015; Urlep et al., 2020). The network evidence plot was presented in Fig. 2B. The combined data showed good consistency between direct and indirect comparisons (IF = 3.08, 95% CI [0–7.34]) (Fig. S5). Based on the SUCRA values (Fig. S6), pediatric CD exhibited better tolerance to PEN+CDED (SUCRA 88.0%), followed by the PEN (SUCRA 47.0%), PEN+AID-CD (SUCRA 41.4%). Children exhibited the poorest tolerance to EEN (SUCRA 16.3%). The comparative forest plot was presented in Fig. S7. Table 2 summarized the NMA comparison results for the tolerance, indicating that patients demonstrated significantly better tolerance to PEN+CDED (OR = 0.07, 95% CI [0.01–0.61]) compared to EEN.

Table 2 Network meta-analysis comparisons for tolerance.

PEN+CDED						
0.39 (0.02, 8.42)	Corticosteroids					
0.17 (0.01, 2.93)	0.43 (0.02, 7.93)	PEN				
0.14 (0.01, 4.06)	0.37 (0.01, 10.89)	0.86 (0.04, 20.93)	PEN+AID-CD			
0.13 (0.01, 2.10)	0.34 (0.02, 5.68)	0.78 (0.08, 7.93)	0.91 (0.04, 20.51)	Infliximab		
0.07 (0.01, 0.61)	0.18 (0.02, 1.68)	0.43 (0.06, 2.88)	0.50 (0.04, 6.50)	0.55 (0.09, 3.20)	EEN	
Notes:

ORs higher than 1 favour the row-defining treatment.

ORs lower than 1 favour the column-defining treatment.

Significant results are in bold.

EEN, exclusive enteral nutrition; PEN, partial enteral nutrition; AID-CD, anti-inflammatory diet for Crohn’s disease; CDED, Crohn’s disease exclusion diet.

Secondary outcome

C-reactive protein

A total of 9 RCTs with 282 patients reported the changes in C-reactive protein (CRP) levels before and after treatment. Among them, 214 patients received dietary therapy strategies (Levine et al., 2019; Kierkuś et al., 2013; Terrin et al., 2002; Pigneur et al., 2019; Hart et al., 2020; Borrelli et al., 2006; Johnson et al., 2006; Urlep et al., 2020; Ruuska et al., 1994). The network evidence plot was shown in Fig. 2C. As no closed loops were formed in the network plot, we did not test for inconsistency in the NMA and therefore only employed a consistency model. Based on the SUCRA values (Fig. S8), in terms of reducing CRP levels, PEN+AID-CD was the most effective intervention (SUCRA 67.9%), followed by EEN (SUCRA 58.8%) and PEN+CDED (SUCRA 48.6%). PEN showed the least effectiveness (SUCRA 36.0%). The comparative forest plot was presented in Fig. S9. Table 3 summarized the NMA comparison results for changes in CRP levels, indicating no significant differences among the various dietary therapy strategies.

Table 3 Network meta-analysis comparisons for CRP.

PEN+AID-CD					
−2.80 (−16.57, 10.97)	EEN				
−4.10 (−29.04, 20.84)	−1.30 (−22.09, 19.49)	PEN+CDED			
−4.48 (−18.78, 9.82)	−1.68 (−5.55, 2.19)	−0.38 (−21.53, 20.76)	Corticosteroids		
−9.80 (−49.01, 29.41)	−7.00 (−43.72, 29.72)	−5.70 (−47.89, 36.49)	−5.32 (−42.24, 31.60)	PEN	
Notes:

MDs lower than 0 favour the column-defining treatment.

MDs higher than 0 favour the row-defining treatment.

Significant results are in bold.

EEN, exclusive enteral nutrition; PEN, partial enteral nutrition; AID-CD, anti-inflammatory diet for Crohn’s disease; CDED, Crohn’s disease exclusion diet.

Albumin

A total of 10 RCTs, involving 289 patients, reported the changes in serum albumin levels before and after treatment, with a total of 182 patients receiving dietary therapy strategies (Kierkuś et al., 2013; Terrin et al., 2002; Luo et al., 2017; Pigneur et al., 2019; Papadopoulou et al., 1995; Hart et al., 2020; Borrelli et al., 2006; Johnson et al., 2006; Urlep et al., 2020; Ruuska et al., 1994). The network evidence plot was presented in Fig. 2D. As no closed loops were observed in each network plot, inconsistency in the NMA was not tested, and a consistency model was chosen. The comparison-adjusted funnel plot indicated no substantial evidence of small-study effects among the included studies (Fig. S12B). Based on the SUCRA values (Fig. S10), EEN was the most effective dietary therapy strategies (SUCRA 74.5%) for increasing serum albumin levels, followed by PEN+AID-CD (SUCRA 72.8%), while PEN exhibited the least effectiveness (SUCRA 14.3%). The comparative forest plot could be found in Fig. S11. Table 4 summarized the NMA comparison results for changes in albumin levels, demonstrating that EEN (MD = 2.23, 95% CI [0.66–3.81]) yielded significantly superior outcomes in improving serum albumin levels compared to corticosteroids.

Table 4 Network meta-analysis comparisons for albumin.

EEN					
−0.30 (−4.86, 4.26)	PEN+AID-CD				
0.43 (−4.19, 5.06)	0.73 (−5.76, 7.23)	Infliximab			
2.23 (0.66, 3.81)	2.53 (−2.29, 7.36)	1.80 (−3.01, 6.61)	Corticosteroids		
3.30 (−0.02, 6.62)	3.60 (−2.04, 9.24)	2.87 (−2.83, 8.56)	1.07 (−2.61, 4.74)	PEN	
Notes:

MDs higher than 0 favour the column-defining treatment.

MDs lower than 0 favour the row-defining treatment.

Significant results are in bold.

EEN, exclusive enteral nutrition; PEN, partial enteral nutrition; AID-CD, anti-inflammatory diet for Crohn’s disease.

Fecal calprotectin

A total of 4 RCTs with 196 patients reported the changes in fecal calprotectin (FC) levels before and after treatment. Among them, 143 patients received dietary therapy strategies (Levine et al., 2019; Hart et al., 2020; Lee et al., 2015; Urlep et al., 2020). The network evidence plot was shown in Fig. 2E. Since the closed loop in the network plot consists of only one multi-arm study, inconsistency in the NMA was not tested, and a consistency model was chosen. Based on the SUCRA values (Fig. S13), in terms of reducing FC levels, PEN+AID-CD was the most effective intervention (SUCRA 78.3%), followed by PEN+CDED (SUCRA 62.4%) and EEN (SUCRA 50.4%). PEN showed the least effectiveness (SUCRA 16.5%). The comparative forest plot was presented in Fig. S14. Table 5 summarized the NMA comparison results for changes in FC levels, indicating that PEN+AID-CD (MD = −114.09, 95% CI [−210.06 to −18.11]) was significantly superior to EEN in improving FC levels.

Table 5 Network meta-analysis comparisons for fecal calprotectin.

PEN+AID-CD						
410.88 (−3,151.78, 3973.55)	PEN+CDED					
−114.09 (−210.06, −18.11)	−524.97 (−4,086.34, 3,036.41)	EEN				
−174.08 (−560.24, 212.07)	−584.97 (−4,165.93, 2,995.99)	−60.00 (−434.04, 314.04)	Infliximab			
−176.99 (−732.54, 378.56)	−587.87 (−4,191.04, 3,015.30)	−62.90 (−610.17, 484.36)	−2.91 (−665.75, 659.94)	Corticosteroids		
−416.08 (−898.37, 66.20)	−826.97 (−4,419.56, 2,765.63)	−302.00 (−774.64, 170.65)	−242.00 (−636.41, 152.41)	−239.09 (−962.18, 483.99)	PEN	
Notes:

MDs lower than 0 favour the column-defining treatment.

MDs higher than 0 favour the row-defining treatment.

Significant results are in bold.

EEN, exclusive enteral nutrition; PEN, partial enteral nutrition; AID-CD, anti-inflammatory diet for Crohn’s disease; CDED, Crohn’s disease exclusion diet.

Safety

Dietary therapy strategies are widely recognized for their safety advantages due to neither triggering immune suppression nor systemic toxic reactions (Ruemmele et al., 2014). In the study by Levine et al. (2019), detailed reports of adverse events were provided, with 12 cases in the PEN+CDED group (including transient ALT elevation, viral infection, hospitalization-fever and gastroenteritis, flare, hospitalization flare, elevated amylase/lipase, upper respiratory infection, hypophosphatemia, herpes zoster, acute gastroenteritis, drug-induced emesis, hematemesis) and 13 cases in the EEN group (including transient ALT elevation, exacerbation, Nausea, flare, hospitalization flare, diarrhea, cough, skin abscess, headache). The proportion of diet-related adverse events was higher in the EEN group compared to the PEN+CDED group (EEN: 4/13, 30.7%; PEN+CDED: 2/12, 16.6%), and one severe adverse event requiring hospitalization occurred in the PEN+CDED group. Other studies reported rare adverse events, such as vomiting related to taste in enteral nutrition.

Discussion

It is the first systematic review and network meta-analysis to assess dietary therapy strategies for pediatric CD. The comprehensive analysis revealed that PEN+CDED, EEN, and PEN+AID-CD appeared to be better than PEN for inducing remission in the included trials. Moreover, PEN+CDED showed significantly better tolerance compared to EEN, while PEN+AID-CD was significantly more effective than EEN in improving FC levels. Based on SUCRA, PEN+CDED was likely the best modality for inducing remission and tolerance. PEN+AID-CD was likely the ranks first in improving CRP and FC levels, and EEN was likely the ranks first in improving albumin levels. It is noteworthy that, except for tolerance, PEN consistently ranks lowest in all evaluated aspects. Different dietary therapy strategies have their own advantages, and different dietary therapy strategies should be formulated for pediatric patients with CD from different backgrounds.

The increasing prevalence of pediatric CD worldwide is associated with the westernization diets, which alter gut microbiota and disrupt intestinal immune function. Therefore, employing dietary strategies has emerged as a frontline therapy for treating pediatric CD (Ruemmele et al., 2014). EEN, a nutrient-dense formula diet without solid foods, is the first-line choice for inducing remission in mild-to-moderate CD (Caruso, Lo & Núñez, 2020). Our results confirm the results of the prior meta-analysis that induced remission rate of EEN is similar to that of corticosteroids (Yu, Chen & Chen, 2019). Previous studies had shown that EEN could improve disease activity, inflammatory markers, and growth retardation in pediatric CD (Rubio et al., 2011; Cameron et al., 2013). Moreover, our data highlight EEN’s superiority in enhancing albumin levels, which indicated the superiority of EEN in the treatment of severely malnourished pediatric CD (Wall, Day & Gearry, 2013). A recent large retrospective study involving over 700 children showed that the use of EEN in pediatric CD is associated with a lower long-term risk of corticosteroids dependency and hospitalization compared to corticosteroids (Plotkin et al., 2024). These results may support dietary therapy for pediatric CD. However, this dietary therapy strategy often proves challenging to sustain over the long term due to the burden it places on families. Thus, exploring different dietary therapy strategies and improving tolerance are critical research directions in diet therapy. One potential solution to enhance treatment compliance is the introduction of small amounts of food (Gray et al., 2012; Sohouli et al., 2022; Rodrigues et al., 2007).

Based on previous studies published, PEN seem to be better tolerated than EEN (Gray et al., 2012). However, in clinical practice, PEN is generally not recommended to induce remission. Existing studies of PEN have found that while PEN may provide some modest benefit, the clinical response in terms of inducing remission is not comparable to that of EEN (Lee et al., 2015; Brückner et al., 2020). Our study showed that PEN might be better tolerated than EEN in pediatric CD according to SUCRA, but consideration was needed when selecting PEN for inducing remission in children with mild to moderate CD. The inclusion of an unrestricted free diet might expose children to potentially harmful dietary antigens, compromising the anti-inflammatory effects of enteral nutrition (Sigall Boneh et al., 2017). Consequently, researchers are considering combining PEN with ED to improve dietary therapy.

PEN+CDED, designed by Arie Levine’s team at Wolfson Medical Center in Israel (Levine et al., 2019). Levine et al. (2019) noted that compared to EEN, tolerance to CDED + PEN was significantly better. Moreover, the clinical remission rate did not differ significantly between CDED + PEN and EEN, indicating that CDED + PEN and EEN were similarly effective at inducing pediatric CD remission. A real-world data study indicated that PEN+CDED has comparable efficacy to EEN in inducing remission in pediatric CD patients and can result in better weight gain (Niseteo et al., 2022). Similar to EEN, PEN+CDED may work by eliminating harmful dietary components affecting intestinal permeability and gut microbiota. The diet focuses on natural foods while avoiding processed and allergenic ones (Levine et al., 2019). Although no statistically significant differences were observed between EEN and PEN+CDED, the present study found that PEN+CDED had the highest probability of being the best dietary therapy strategies in inducing clinical remission with a SUCRA of 90.5%. Moreover, our NMA revealed that PEN+CDED exhibited better tolerability compared to EEN, which is consistent with previous study (González-Torres et al., 2022; Levine et al., 2019). Enhancing tolerance not only promotes therapeutic outcomes but also alleviates the psychological impact of the disease on them. Szczubelek et al. (2021) used the Inflammatory Bowel Disease Questionnaire to measure the impact of CDED on the quality of life of CD patients and found that after 6 and 12 weeks of intervention, the patients’ quality of life improved significantly when compared with baseline. The study indicated that during the maintenance induction remission phase, reducing to 25% PEN for energy requirements combined with CDED can still continuously decrease patients’ FC levels (Levine et al., 2019). At week 12, the proportion of patients maintaining steroid-free remission was significantly higher with CDED and PEN+CDED compared to EEN (Levine et al., 2019). This is because maintaining remission requires the exclusion of harmful components found in regular diets, and re-exposure to food antigens can lead to inflammation rebound and a reduction in sustained remission. Overall, PEN+CDED facilitates the development of a long-term sustainable and healthy lifestyle for patients and can be effective in overcoming the difficulties associated with EEN, as it allows access to food in a standardized and controlled manner, and can be used not only as an induction strategy, but also as a good option for long-term maintenance of remission.

PEN+AID-CD represents a distinct dietary therapy strategies that combines PEN with AID-CD. Similar to CDED, AID-CD excludes processed foods with additives, animal fat, sugar, dairy, and gluten. AID-CD emphasizes Slovenian local recipes, prohibiting fried foods and using locally sourced ingredients (Urlep et al., 2020). Although AID-CD may seem stricter, it does not necessarily result in better therapeutic outcomes. Patient compliance may decline due to the higher enteral nutrition proportion in PEN+AID-CD compared to PEN+CDED. Notably, PEN+AID-CD significantly outperformed EEN in reducing FC levels. Based on SUCRA, PEN+AID-CD may be superior to other dietary therapy strategies in improving CRP and FC levels. Both FC and CRP can assess inflammatory status; FC is more sensitive than CRP in reflecting disease activity in CD. This may be due to the prohibition of fried foods in the AID-CD. Research shows that fried foods rich in saturated fat and trans fat can trigger oxidative stress and pro-inflammatory responses, impacting health (Keewan et al., 2020).

Regarding safety, vomiting is the most frequently reported adverse reaction associated with EEN, due to the taste of the polymeric formula. PEN+CDED has a lower incidence of adverse events compared to EEN, while one study mentioned a severe adverse event associated with PEN+CDED, no other studies reported any serious adverse events (Levine et al., 2019).

More researchers are currently focusing on the impact of dietary therapy strategies on pediatric CD (El Amrousy et al., 2022; Suskind et al., 2020). Additionally, certain studies on exclusionary whole food diets have garnered our attention. Strisciuglio et al. (2020) conducted a one-year cross-sectional study involving 53 pediatric patients with CD, indicating that adherence to a Mediterranean diet significantly reduces FC levels in children with CD. A retrospective study examined the long-term outcomes of 20 pediatric patients with active CD following a specific carbohydrate diet. The patients exhibited sustained improvement in their PCDAI (Obih et al., 2016). However, existing studies do not provide sufficient evidence to support the use of exclusionary whole food diets for inducing remission in pediatric CD patients, making it difficult for patients and physicians to choose an appropriate diet. We believe that there is still a need for larger scale direct comparative studies on dietary therapy strategies and “gold standard” treatments in the future, rather than placebos. Meanwhile, the inherent heterogeneity of table food is a challenge that such research needs to face. It should be noted that the commonality of these diets is the limitation of certain dietary components, as these limitations on oneself, without supplementation, are incomplete in nutrition, and it is important for nutritionists to participate. This is particularly important for growing children or adolescents, monitored by trained professionals.

However, it is important to acknowledge certain limitations inherent in our NMA. Firstly, there is only one RCT on PEN+AID-CD and PEN+CDED, respectively. thus, this may have affected the power and precision of the results. Regarding this, we found ongoing RCTs on clinicaltrials.gov investigating dietary therapy strategies for inducing remission in CD (NCT04239248, NCT05284136). These studies are in the recruitment phase, and we will diligently follow their progress to provide future updates for our NMA. Secondly, it is acknowledged that NMA typically relies on the inclusion of appropriately homogeneous trials. However, to include a larger number of studies, we did not extract data at the same time points. Finally, only English-language databases were searched; although we tried to collect the relevant evidence as comprehensively as possible, there may be eligible studies not indexed in English language databases.

Conclusion

This systematic review and network meta-analysis provides evidence of better reported effectiveness with PEN+CDED, EEN, and PEN+AID-CD compared to PEN in including clinical remission. PEN+CDED was likely to be the best intervention to induce remission and improve tolerance. For pediatric CD patients with hypoalbuminemia and other malnutrition, we may recommend the application of EEN. For pediatric CD patients with high inflammation, we may recommend the PEN+AID-CD. All dietary therapies strategies have high safety. More well-designed RCTs are still needed to explore the efficacy of various dietary strategies in the future.

Supplemental Information

Supplemental Information 1 PRISMA checklist.

Supplemental Information 2 Supplementary Figures.

Additional Information and Declarations

Competing Interests

Author Contributions

Data Availability

The authors declare that they have no competing interests.

Jiaze Ma conceived and designed the experiments, performed the experiments, authored or reviewed drafts of the article, and approved the final draft.

Jinchen Chong performed the experiments, authored or reviewed drafts of the article, and approved the final draft.

Zhengxi Qiu analyzed the data, prepared figures and/or tables, and approved the final draft.

Yuji Wang analyzed the data, prepared figures and/or tables, and approved the final draft.

Tuo Chen conceived and designed the experiments, authored or reviewed drafts of the article, and approved the final draft.

Yugen Chen conceived and designed the experiments, authored or reviewed drafts of the article, and approved the final draft.

The following information was supplied regarding data availability:

This is a systematic review and meta-analysis.

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
