# Peer review of "Efficacy of different dietary therapy strategies in active pediatric Crohn’s disease: a systematic review and network meta-analysis"

_PeerJ, doi:10.7717/peerj.18692_

## Round 0.1 · original submission · Minor Revisions

1. Please provide justification for the 4-week minimum intervention duration chosen for RCTs.
2. Add details about data quality control and validation procedures.
3. Clarify how disease staging differences were handled across studies.
4. Explain methods used to address potential confounding factors.
5. Add more detailed figure legends.
6. Clarify the connection between primary and secondary outcomes.
7. Enhance presentation of SUCRA values interpretation.
8. Add details about data normalization procedures and Include information about handling heterogeneity.
9. Please check grammar errors thoroughly. (e.g., "of to Fig out")

Reviewer 1 ·

Basic reporting

• Figure Quality: The manuscript's figures suffer from poor resolution and small font sizes, making data interpretation challenging. Enhancing figure clarity would improve readability.
• Language and Terminology: While the manuscript is generally well-written, some sections could benefit from clearer phrasing. Consistent terminology is recommended, particularly when referring to dietary therapy strategies and intervention specifics.

Experimental design

• RCT Justification: The manuscript includes RCTs with a minimum intervention duration of four weeks; however, further justification on why this duration is optimal for an induction phase in pediatric Crohn’s Disease would be beneficial. A more detailed rationale for this choice would support the study’s design strength.
• Data Quality Control: There is no detailed mention of data quality control strategies. Given the potential variance across studies, particularly in pediatric populations, clarifying any normalization, data validation, or consistency checks used would strengthen the methodology.
• Search Terms and Disease Differentiation: The search strategy combines "inflammatory bowel diseases" and "Crohn's disease" with "OR," potentially introducing heterogeneity by including ulcerative colitis cases under the broader IBD umbrella. Additional criteria to distinguish between Crohn's Disease and ulcerative colitis would improve the specificity and applicability of the results.
• Staging of Crohn’s Disease and Population Balance: The manuscript does not describe methods for addressing differences in Crohn’s Disease stages across patients or balancing treatment groups accordingly. Such variability may confound the results, especially given the potential for disease progression over large time gaps. A more thorough discussion on how these confounding factors were managed or mitigated would improve the robustness of the findings.

Validity of the findings

• Endpoint and Remission Duration: Although remission rate is a meaningful endpoint, it is limited in scope for assessing long-term management efficacy. The study does not explore remission duration or factors affecting sustained remission, which are critical in chronic disease management. Including a discussion or data analysis on remission duration, if available, would enhance the relevance of the findings for clinical application.

Additional comments

The manuscript addresses an important topic, comparing dietary therapy strategies in pediatric Crohn's Disease. The study design and statistical analyses are generally sound, yet the lack of detailed justification for certain experimental parameters (e.g., intervention duration, quality control strategies) weakens the overall rigor. Figures need improvement for clearer presentation, and the exploration of sustained remission as an endpoint would add substantial value. Addressing these points could enhance both the scientific robustness and clinical applicability of the findings.

·

Basic reporting

This is an interesting study, in which the authors focused attention on the effectiveness and tolerance of different dietary therapy strategies for the Crohn's disease (CD) in pediatric patients. Ma et al. found that the partial enteral nutrition plus Crohn's disease exclusion diet (PEN+CDED) exhibited significant superiority over PEN alone and higher tolerance than EEN. They further revealed that the PEN+CDED intervention (90.5%) achieved the highest ranking in clinical remission rate. This systematic review provides evidence that PEN+CDED can be used as an alternative treatment to Exclusive enteral nutrition and is more suitable for long-term management in children. However, there are some details that need to be improved. Following is, the content to be supplemented in the revised paper.

Experimental design

In the introduction and discussion parts, what is the connection between the primary results and secondary outcome? It will be a little better if you add the connection between the primary results and secondary outcome in the article.

Validity of the findings

This systematic review provides evidence that PEN+CDED can be used as an alternative treatment to Exclusive enteral nutrition and is more suitable for long-term management in children.

Additional comments

Line109-110. What does “of to Fig out” mean?

The quality of the Figure 1 is blurred. Please renew the Figure.

Line 126-127. It was unclear what the authors wanted to express. Could you correct it, please?

---

## Round 0.2 · accepted · Accept

Authors have fully addressed all comments. I think this paper can be accepted for publication.

Reviewer 1 ·

Basic reporting

No more comments.

Experimental design

No more comments.

Validity of the findings

No more comments.

Additional comments

The authors have thoroughly addressed all my concerns and made significant improvements to the manuscript. I now recommend its acceptance.